# Immune and sex-biased gene expression in the threatened Mojave desert tortoise, *Gopherus agassizii*

Cindy Xu[1], Greer A. Dolby[1,2], K. Kristina Drake[3]*, Todd C. Esque[3], Kenro Kusumi[1]*

**1** School of Life Sciences, Arizona State University, Tempe, Arizona, United States of America, **2** Center for Mechanisms of Evolution, Biodesign Institute, Arizona State University, Tempe, Arizona, United States of America, **3** Western Ecological Research Center, U.S. Geological Survey, Henderson, Nevada, United States of America

* Kdrake@usgs.org (KKD); kenro.kusumi@asu.edu (KK)

**Data Availability Statement:** Sequence data is available via NCBI SRA accession numbers SRS3600670 to SRS3600682, BioSample accession numbers SAMN09727116 to SAMN09727140, and BioProject accession

## Abstract

The immune system of ectotherms, particularly non-avian reptiles, remains poorly characterized regarding the genes involved in immune function, and their function in wild populations. We used RNA-Seq to explore the systemic response of Mojave desert tortoise (*Gopherus agassizii*) gene expression to three levels of *Mycoplasma* infection to better understand the host response to this bacterial pathogen. We found over an order of magnitude more genes differentially expressed between male and female tortoises (1,037 genes) than differentially expressed among immune groups (40 genes). There were 8 genes differentially expressed among both variables that can be considered sex-biased immune genes in this tortoise. Among experimental immune groups we find enriched GO biological processes for cysteine catabolism, regulation of type 1 interferon production, and regulation of cytokine production involved in immune response. Sex-biased transcription involves iron ion transport, iron ion homeostasis, and regulation of interferon-beta production to be enriched. More detailed work is needed to assess the seasonal response of the candidate genes found here. How seasonal fluctuation of testosterone and corticosterone modulate the immunosuppression of males and their susceptibility to *Mycoplasma* infection also warrants further investigation, as well as the importance of iron in the immune function and sex-biased differences of this species. Finally, future transcriptional studies should avoid drawing blood from tortoises via subcarapacial venipuncture as the variable aspiration of lymphatic fluid will confound the differential expression of genes.

## Introduction

The innate and adaptive immune systems of non-avian reptiles remain a challenge to characterize, particularly in how they function relative to avian and mammalian model systems [1]. Challenges to this characterization are fourfold. First, while the number of available reference genome assemblies for non-avian reptiles is increasing (e.g., [2–7]), these resources are newer and less well-developed than those of model systems, such as human, chicken, and African

number PRJNA483175. All other results are available as appendices or bundled in Harvard Dataverse: https://doi.org/10.7910/DVN/XOKGGW.

**Funding:** This research was supported in part by the National Science Foundation EID grant #1216054 (K.K.D. and T.C.E.), U.S. Bureau of Land Management California Desert, California and Las Vegas, Nevada Districts agreement #L11PG00370 (K.K.D. and T.C.E), U.S. Geological Survey, Western Ecological Research Center [GX16ZC00BQAP2, GX16ZC00BQAP4] (G.A.D and K.K.), and Coyote Springs Investment LLC (K.K.D. and T.C.E). Any use of trade, product or firm names in this publication is for descriptive purposes only and does not imply endorsement by the U.S. government.

**Competing interests:** I have read the journal's policy and the authors of this manuscript have the following competing interests: G.A.D. is a member of the Board of Directors for the Desert Tortoise Council, a nonprofit conservation organization that protects desert tortoises and their habitats and funding support was provided by Coyote Springs Investment LLC. The DTC and Coyote Springs Investment LLC had no involvement in the design, implementation, or interpretation of this study. These do not alter our adherence to PLOS ONE policies on sharing data and materials.

clawed frog [8–10]. As a consequence, it is still largely unknown how much of the "immune gene set" of human, chicken, and frog is conserved in non-avian reptiles. Second, there are fewer functional studies about the immunological mechanisms and pathways governing how non-avian reptiles respond to pathogens compared to mammal and bird systems [1]. Third, the closest-related functional models—human and chicken—are endotherms so their immune processes occur at a consistent internal temperature unlike non-avian reptiles whose ectothermy means they have a wider set of physiological and activity states. The biochemistry that mediates immune responses may range in efficiency under these different conditions, adding a layer of complexity to their immune function [11,12]. Finally, how the sex of an organism modulates immune functions is now well-documented in mammals [13], but how sex-biased differences manifest in the immune function of non-avian reptiles with temperature-dependent sex determination is less known [14].

In many species, males and females exhibit differences in anatomy, morphology, physiology, and behavior. To some extent, sexually dimorphic traits are the product of sex differences in gene expression, which allow phenotypic differences from a common autosomal genome. Genes differentially expressed by sex are known as sex-biased genes and can be female or male biased, depending on which sex exhibits higher levels of transcription [15,16]. Sex-biased gene expression is the result of differential gene regulation between males and females, but in taxa with specialized sex chromosomes, gene dosage also plays a role in unequal transcription. Sex affects immune function as well as disease prevalence and severity. In general, females mount stronger innate and adaptive immune responses than males [17,18]. Possible explanations for this are differential regulation of gene expression by sex hormones [19,20] or differences in behavior [17,21]. Many studies focus on sex-biased expression of immune genes on sex chromosomes, but the majority of immune genes are located on autosomal chromosomes and these warrant further study [22]. How temperature-based sex determination affects sex-biased gene expression is also understudied. Therefore, the role of environment on disease susceptibility and prevalence, as well as how these effects manifest differently by sex are topics of considerable interest that have broad implications for the management of wildlife.

One way to fill in these knowledge gaps is to assay the transcriptional response of how a non-avian reptile responds to infection where the host-pathogen relationship is reasonably well understood. Infection and disease of the Mojave desert tortoise (*Gopherus agassizii*) by the pathogenic bacteria *Mycoplasma* spp. is among the most extensively characterized in chelonians [23]. Furthermore, because desert tortoises are long-lived animals, the cumulative effects of exposure to stressors such as prolonged infection may have considerable long-term impact for these animals. Knowledge gained about transcriptomic response to infection in this system will shed light on the innate and adaptive immune response of non-avian reptiles to infection broadly, including sex-biased effects.

Learning about this host-pathogen relationship is a major conservation priority for the desert tortoise, which was listed as threatened under the US Endangered Species Act in 1990 [24]. In this host-pathogen relationship, the *Mycoplasma agassizii* bacteria disrupt the tissue of the ciliated mucosal epithelium leading to upper respiratory tract disease (URTD). Although this disease has been extensively studied in desert tortoises, it remains unclear if or how *M. agassizii* bind to epithelial cells or if they just reside in the mucosal layer similar to other respiratory microbes [25]. In desert tortoises, URTD can lead to severe damage to the tissues of the upper respiratory tract and occlusion of nasal passages by thick mucosal discharges. The URTD is also thought to be a direct cause of mortality in Mojave desert tortoises [23,26–28].

Higher pathogen loads from *M. agassizii* generally correlate with increased clinical signs and adaptive antibody responses [29,30]; however, antibody production can be delayed in this species by months to years after *M. agassizii* infection [30,31]. By the time most tortoises are

classified has having URTD and an antibody response is detected, individuals have likely been infected for a long period of time. For some time initially after infection by pathogen, tortoises may not show clinical signs or an antibody response but would test positive for *M. agassizii* by qPCR [32–34]. As many of the recovery unit populations remain below viability levels [35], it is important to the health and viability of the species to understand how these bacterial infections and URTD impact the health of tortoises.

To learn more about the transcriptional response to this host-pathogen relationship, we used RNA-Seq to analyze the blood-based gene expression patterns of male and female tortoises with severe *M. agassizii* infection, tortoises inoculated with *M. agassizii*, and uninfected tortoises. Based on previous studies, we expect cytokines (e.g. IFN-γ, interleukins, TNFα1) and inflammatory signaling pathway genes to be expressed at higher levels in tortoises with bacterial infection, as they are important mediators of a host's defense to pathogens. Since the tortoises in this study were infected for long periods of time, we also expect signs of chronic infection and humoral immune response such as immunoglobulins or lymphocyte-specific cluster of differentiation genes, which are necessary for targeting, activation, and survival.

## Materials and methods

### Experimental design

We analyzed 25 tortoise individuals across three experimental groups with varying degrees of bacterial infection including: **1)** individuals discovered with severe *M. agassizii* infection and were not inoculated from experimentation (Severe Infection group, SI); **2)** individuals inoculated with *M. agassizii* showing medium infection (Medium Infection group, MI); and **3)** individuals serving as a control without any known infection (No Infection group, NI; Table 1). In our study, both medium and severe infection groups were sampled from captive colonies with documented infection for multiple years. We were not able to incorporate subclinical animals with early or low levels of infection in this study. All handling and experiments using animals were approved by the U.S. Geological Survey-Western Ecological Research Center Animal Care and Use Committee (WERC 2012–03) and covered under state (Nevada Division of Wildlife Permit #S33762) and federal (U. S. Fish and Wildlife Service TE-030659) permits.

For the severe infection (SI) group, we chose captive adults (N = 9; 6F:3M) from the Desert Tortoise Conservation Center (DTCC) in Clark County, Nevada, USA (35.975256, -115.251048) that were classified with severe infection based on long-term health evaluations by experienced veterinarians. Each tortoise in this category had a confirmed long-term *M. agassizii* infection and multiple clinical signs of potential illnesses associated with long-term weight loss and reduced or under-conditioned body condition (Table 1). Due to their poor overall health, consistent with captive herd management protocols and based on veterinary guidance, most tortoises (8 of 9) were euthanized following sample collection and immediately necropsied to evaluate tissue conditions morphologically and histologically. Tortoises were euthanized after this study period by licensed veterinarians using a mixture of ketamine (5 mg/kg) and dexmedotomidine (0.1 mg/kg) injected intramuscularly as an anesthetic. Once animals were non-responsive, Euthasol (2 ml/kg) was injected intravenously into the subcarapacial cranial plexus [37].

For the medium infection group, we also used captive adult tortoises (N = 7; 1F:6M) from the DTCC that were experimentally exposed to *M. agassizii* as part of a previous study [29]. These tortoises tested positive for the presence of *M. agassizii* bacteria for four years and exhibited targeted immune responses (specific antibody production measured using enzyme-linked immunosorbent assay (ELISA) tests) to *M. agassizii* as well as intermittent clinical signs associated with inflammatory responses to this infection for two years prior to sampling [31]. For the control group we chose clinically normal, adult tortoises (N = 9; 5F:4M) from a wild

**Table 1. Clinical condition of adult captive and wild tortoises.**

| Tortoise ID | ID | Sex | MCL (mm) | Mass (g) | Eyes | Nares | Oral Cavity | Skin | Shell | Other | Body Condition Score |
|---|---|---|---|---|---|---|---|---|---|---|---|
| CS0004 | NI-1 | F | 244 | 2370 | – | – | – | – | – | – | 6 |
| CS0005 | NI-2 | M | 266 | 3401 | R | – | – | – | – | – | 4 |
| CS0011 | NI-3 | M | 281 | 4705 | R | – | – | – | – | – | 5 |
| CS0023 | NI-4 | F | 250 | 2810 | R | – | – | – | – | – | 6 |
| CS0049 | NI-5 | F | 260 | 2540 | R | – | – | – | – | – | 5 |
| CS0052 | NI-6 | M | 277 | 4280 | R | – | – | – | – | – | 4 |
| CS0072 | NI-7 | F | 261 | 3300 | R | – | – | – | – | – | 4 |
| CS0078 | NI-8 | F | 268 | 3440 | R | – | – | – | – | – | 4 |
| CS0083 | NI-9 | M | 312 | 5560 | R | – | – | – | – | – | 4 |
| 15780 | MI-1 | M | 244 | 2718 | DS, E | DS, Er | – | – | – | – | 4 |
| 21804 | MI-2 | M | 245 | 2794 | E | DS, O | – | – | – | – | 5 |
| 22003 | MI-3 | M | 274 | NA | E | DS, Er | – | – | – | – | 4 |
| 22314 | MI-4 | M | 238 | 3016 | R | DS | – | – | – | – | 4 |
| 22335 | MI-5 | F | 256 | 3056 | E | – | – | – | – | LR | 4 |
| 22390 | MI-6 | M | 238 | 2840 | R | – | – | – | – | – | 4 |
| 22399 | MI-7 | M | 265 | 3390 | E | – | – | – | – | LR | 4 |
| 18518 | SI-1 | M | 274 | 2165 | R | A, DS, Er | – | – | – | CM | 3 |
| 18602 | SI-2 | F | 281 | 4329 | DS, CR, E, R | A, DS, Er | CP | – | CD | – | 4 |
| 18619 | SI-3 | F | 260 | 3037 | E, DM, CR | A, DS, DM, Er, O | CP | – | – | CM, LR | 4 |
| 18789 | SI-4 | F | 250 | 2430 | DS, E | A, DS, DM, Er, O | – | – | – | – | 4 |
| 19156 | SI-5 | F | 274 | 3300 | DS, E, CR, | A, DM, O | – | – | – | – | 4 |
| 19431 | SI-6 | M | NA | 3140 | E, R | Er, O | – | – | – | – | 4 |
| 19392 | SI-7 | F | 287 | 4028 | E | DS, Er | – | – | – | W | 4 |
| 19730 | SI-8 | F | 275 | 3786 | E, ER | A, DS, DM, Er | – | – | – | – | 4 |
| 21042 | SI-9 | M | 273 | 3396 | E, CR, R | DM, DS, Er | – | – | – | – | 5 |

The clinical condition of adult captive tortoises with medium (M1-M7; 1F:6M) and severe (SI1-SI9; 6F:3M) infection as well as wild tortoises with no infection (NI1-NI9; 5F:4M) in Clark County, Nevada, USA. Tortoises were evaluated mid-summer (July) immediately before sampling of blood. The following codes indicate clinical anomalies observed during evaluation: A = asymmetrical, CD = cutaneous dyskeratosis, CM = coelomic mass, CP = coloration pale, CR = coloration red, DS = discharge serous, DM = discharge mucoid, E = edema, Er = eroded, L = lesions present, LR = labored respiration, O = occluded, R = recessed, W = weak/lethargic "–" = clinically normal. (MCL = Maximum Carapace Length). Numerical body condition scores (BCSs) were used to assess overall muscle condition and fat stores with respect to skeletal features of the head and limbs [36]. BCS scores were categorized as 'under (1–3),' 'adequate (4–6)' or 'over (7–9)' condition.

population that has been monitored since 2006 in Hidden Valley, Clark County, Nevada, USA (36.528008, -114.975905). Tortoises in this control group were clinically normal based on visual examination by veterinarians and tortoise biologists, and each tortoise had been evaluated and assessed as clinically free of *M. agassizii* infection for 11 consecutive years [38,39] prior to collecting samples for this study. Wild tortoises were not euthanized.

All tortoises were assessed and sampled in peak summer (July–early August) between 0500–0800 hours to minimize circadian and seasonal influences on measured blood analytes. Due to logistical constraints tortoises were sampled during the same season but in different years; Medium Infection (MI) tortoises were sampled in 2017, Severe Infection (SI) tortoises in 2013, and No Infection (NI) tortoises in 2015.

## Choice of blood and venipuncture site

In nonmammalian vertebrates, whole blood is appropriate for gene expression studies for two reasons. First, white blood cells, which include granulocytes, monocytes, and lymphocytes,

allow for the molecular characterization of the host's systemic response to mycoplasma infection. Second, in reptiles erythrocytes are nucleated and transcriptionally active [40,41] and thrombocytes remain as intact cells instead of producing anuclear platelet cytoplasmic fragments [42]. Genes involved in insulin signaling, electron transport chain, stress, and oxidative response are shared between whole blood and liver [41], making whole blood a non-invasive sample to assess immunological and physiological response to infection.

We extracted ~2.5 mL whole blood from all tortoises via jugular venipuncture [43] using either a 1.91-cm, 25-gauge needle-IV infusion set and 3 mL syringe, or subcarapacial venipuncture [37] using a 3.81-cm, 23-gauge needle and 3 mL syringe. All MI and NI tortoises were sampled using subcarapacial venipuncture while SI tortoises were sampled using both subcarapacial and jugular venipuncture (N = 9, 4 subcarapacial:5 jugular). Syringes were coated in sodium heparin to prevent coagulation and blood was collected from severely infected tortoises prior to euthanasia. Two aliquots of whole blood were made per sample. The first aliquot (0.1–0.5 mL blood) was placed immediately into RNeasy® Animal Protect collection tubes (Qiagen, Valencia, CA) for RNA sequencing and gene expression analysis. The second aliquot of ~1.5 mL whole blood was placed in BD Microtainer® tubes with lithium heparin in order to assay blood counts, hematology, blood chemistry, trace elements, and vitamin A concentration (described in [38]). Samples were stored on wet ice for no more than four hours. We separated plasma from the second aliquot using centrifugation (1318 x $g$ for 10 minutes) and stored in an ultracold freezer (-70˚C) until further processing. Aliquots of plasma (0.01 mL) were screened for antibodies specific to *M. agassizii* using an enzyme-linked immunosorbent assay (ELISA; [44]). Sloughed epithelial cells were also collected using sterile oral swabs and screened for *M. agassizii and M. testudineum* using a quantitative Polymerase Chain Reaction (qPCR) assay as described previously [32].

## RNA extraction and sequencing for RNA-Seq

Total RNA was isolated from aliquots of whole blood with minor modifications to the total RNA isolation protocol. Briefly, whole blood samples were thawed on ice, homogenized in RNA lysis buffer, and aliquoted before extracting with acid phenol chloroform twice. Ethanol (100%) was added to each sample and passed through a glass-fiber filter, which binds RNA before eluting with nuclease-free water (mirVana miRNA Isolation Kit with phenol, Ambion, #AM1560, Carlsbad, California). The RNase-Free DNase Set (Qiagen, #79254, Valencia, CA) and RNeasy MinElute Cleanup Kit (Qiagen, #74204, Valencia, CA) were used to treat total RNA for residual DNA and salt contamination. Extracted total RNAs were sent to the Yale Center for Genomic Analysis (YCGA; West Haven, CT) to generate cDNA poly-A-enriched Illumina libraries that were run on two lanes of the Illumina HiSeq 2500 in High Output mode using 75-bp paired-end reads. To avoid batch effects, libraries for all individuals were split across the two sequencing lanes and then read files from the two lanes were concatenated for each sample.

## Quality control and differential expression

We assessed reads for quality using FastQC v0.11.7 and MultiQC v1.5 [45] and performed read trimming using BBDuk v38.00 [46] with a Q28 Illumina quality score to remove low quality sequences; reads of at least 36 bp were retained. Using STAR v2.5.3a [47], the trimmed paired reads were mapped to the *Gopherus agassizii* 1.0 genome [2] and the gopAga1.0 annotation was converted to GTF format via gffreadv0.10.5 (https://daler.github.io/gffutils/api.html). Uniquely mapped reads were used to obtain gene-wise counts from exon sequences using featureCountsv1.6.1 [48]. Paired reads were counted as fragments using the "-p" flag. Multi-mapping and multi-overlapping reads were excluded by default [48].

To explore sample variance, we used Principal Components Analysis (PCA) of regularized log (rlog) count data generated by DESeq2 v1.20.0 [49] (Bioconductor v3.7), which transforms values onto a $\log_2$ scale and normalizes for differences in sequencing depth. We assessed PCA results for variables of interest (experimental immune group, sex), as well as potentially confounding experimental variables: venipuncture site (subcarapacial vs. jugular), captive vs. wild, and sample collection year. In addition to experimental immune group and sex, PCA showed a result for venipuncture site as well. For this reason, we tested for differentially expressed genes (DEGs) for venipuncture site in addition to the DE analyses for experimental immune group and sex to remove it as a potentially confounding variable.

We identified DEGs with DESeq2 [49–51] after excluding four low coverage samples (see results section). Counts were normalized for library size internally, fitted to a negative binomial distribution, and corrected for multiple testing using the Benjamini-Hochberg method (FDR < 0.05). To control for variance associated with experimental group and sex, we used the multifactorial (two-factor) approach that employs the Wald test when evaluating differential expression. The NI experimental group and males were set as the reference levels for the immune and sex-based analyses, respectively. We did not have adequate sampling to analyze the venipuncture site in the multifactor analysis, so we separately ran a one-factor analysis to compare expression of jugular venipuncture versus subcarapacial venipuncture. This one-factor analysis also used the Wald test. Because venipuncture site is a confounding variable, we removed DEGs associated with venipuncture site from the final list of DEGs obtained from experimental group and sex. All results reported are excluding these venipuncture site-associated genes and are divided into genes that are *only* differentially expressed among experimental immune groups (unique immune genes), *only* differentially expressed among sexes (unique sex-biased genes), and those genes differentially expressed among both experimental immune and sex groups (sex-biased immune genes). Analyses and results were run in the R statistical programming environment (http://www.R-project.com). All heatmaps were generated using rlog transformed, mean-centered counts and genes were clustered by Manhattan distance using the Ward method.

## Enrichment analysis

The unique gene lists for experimental immune group and for sex were individually ranked by adjusted *p*-values (padj <0.05) and tested for functional enrichment of Gene Ontology (GO) and Kyoto Encyclopedia of Genes and Genomes (KEGG) pathways in g:Profiler [52]. Functional profiling of differentially expressed genes were queried against the mouse database, excluding *in silico* curated terms, as an ordered list ranked by their adjusted *p*-values. Significant GO categories (padj < 0.05) were identified using Fisher's one-tailed test corrected for multiple testing using the Benjamini-Hochberg method (FDR<0.05) and hierarchically filtered by best-per-parent (moderate) parameters. We visualized significant GO terms (*p* < 0.05) using Reduce and Visualize Gene Ontology (REVIGO, [53]), which uses *simRel* scores as a measure of semantic similarity. A user-provided threshold value of 0.7 was selected for clustering.

## Data accessibility

Sequence data is available via NCBI SRA accession numbers SRS3600670 to SRS3600682, BioSample accession numbers SAMN09727116 to SAMN09727140, and BioProject accession number PRJNA483175 (https://www.ncbi.nlm.nih.gov/bioproject/?term=PRJNA483175). All other results are available as appendices or bundled in Harvard Dataverse: https://doi.org/10.7910/DVN/XOKGGW.

## Results

### Data generation and processing

All samples were sequenced across both Illumina HiSeq lanes to avoid batch effects. We obtained 153 Gb of sequence for the 25 individuals, averaging 30 million reads per individual (N = 25; 3–53 million reads/individual). Average sequence coverage was 2.6-fold lower for four individuals (two SI, two NI) than other individuals, which was evident in PCA (S1 Fig). Removing the four low-coverage samples resulted in a total of 21 samples with 25–53 million paired reads/individual. We only present results from this high-coverage dataset.

After read trimming, 552 million total reads were retained (11–43 million reads/individual) that ranged in length from 36–75 bp. Using STAR yielded a 91.7% mapping rate to the *Gopherus agassizii* 1.0 reference genome, of which 87.1% of reads were uniquely mapped, 4.1% of reads were multiply mapped, and 0.4% of reads were mapped to too many loci (only uniquely mapped reads were retained; S1 Table). Gene expression levels were quantified as the summation of unique fragments mapped to each exon. Overall, individuals in the male and Medium Infection (MI) groups exhibited the highest gene expression variance (Fig 1). However, six of the seven MI individuals were also male, so it is not possible to distinguish whether high variance is a trait specific to males or to having medium *Mycoplasma spp*. infection.

### Data exploration and differential expression

**Venipuncture site.**   Principal Components Analysis (PCA) showed no obvious patterns for two technical variables that were tested: captive vs. wild animals and collection year, suggesting these technical artifacts in our data are weak or absent (S1 Fig). We did observe a pattern associated with jugular vs subcarapacial venipuncture, which we attribute to the presence of varied amounts of lymph fluid in blood samples collected via subcarapacial venipuncture (S1 Fig). Blood collected via subcarapacial venipuncture can be collected within 1–2 min and without extensive animal handling, making it the preferred collection technique by managing agencies [36]. However, the subcarapacial plexus is proximal to lymphatic vessels, and may result in inadvertent aspiration of lymph when collecting blood [37]; jugular venipuncture does not result in any lymph admixture.

There were 53 total genes differentially expressed between venipuncture sites (S1 Appendix). Of these, 38 were unique to venipuncture analysis and showed some involvement in lymph-associated processes, such as regulation by host of viral transcription, T cell activation, eosinophil migration, and response to bacterium (S2 and S3 Figs; S2 Table). To attempt to mitigate the effects of lymph on our immune and sex-based analyses, we *post hoc* removed the 15 venipuncture-site DEGs that were *also* differentially expressed in the immune and/or sex comparisons. This was done to prevent interpretation of the lymph signal as a blood-based immune or sex-biased signal (all jugular venipuncture individuals were female). We note that there have not been RNA-Seq differential expression studies performed on desert tortoises, making this a novel and important result to account for in the design of future studies.

### Immune and sex-biased results

After removing venipuncture site-associated genes, the multifactor Wald test yielded a total of 40 genes that were uniquely differentially expressed among experimental immune groups (Fig 2, S2A and S2B Appendix). Of these 40 genes, 14 were unique to the MI-NI comparison, 21 DEGs were unique to the SI-NI comparison, and five DEGs were differentially expressed in both comparisons (*ABHD8*, *CDO1*, *RNF125*, cell surface hyaluronidase-like, gopAga1_00017729). The 5 genes differentially expressed in both comparisons were upregulated relative

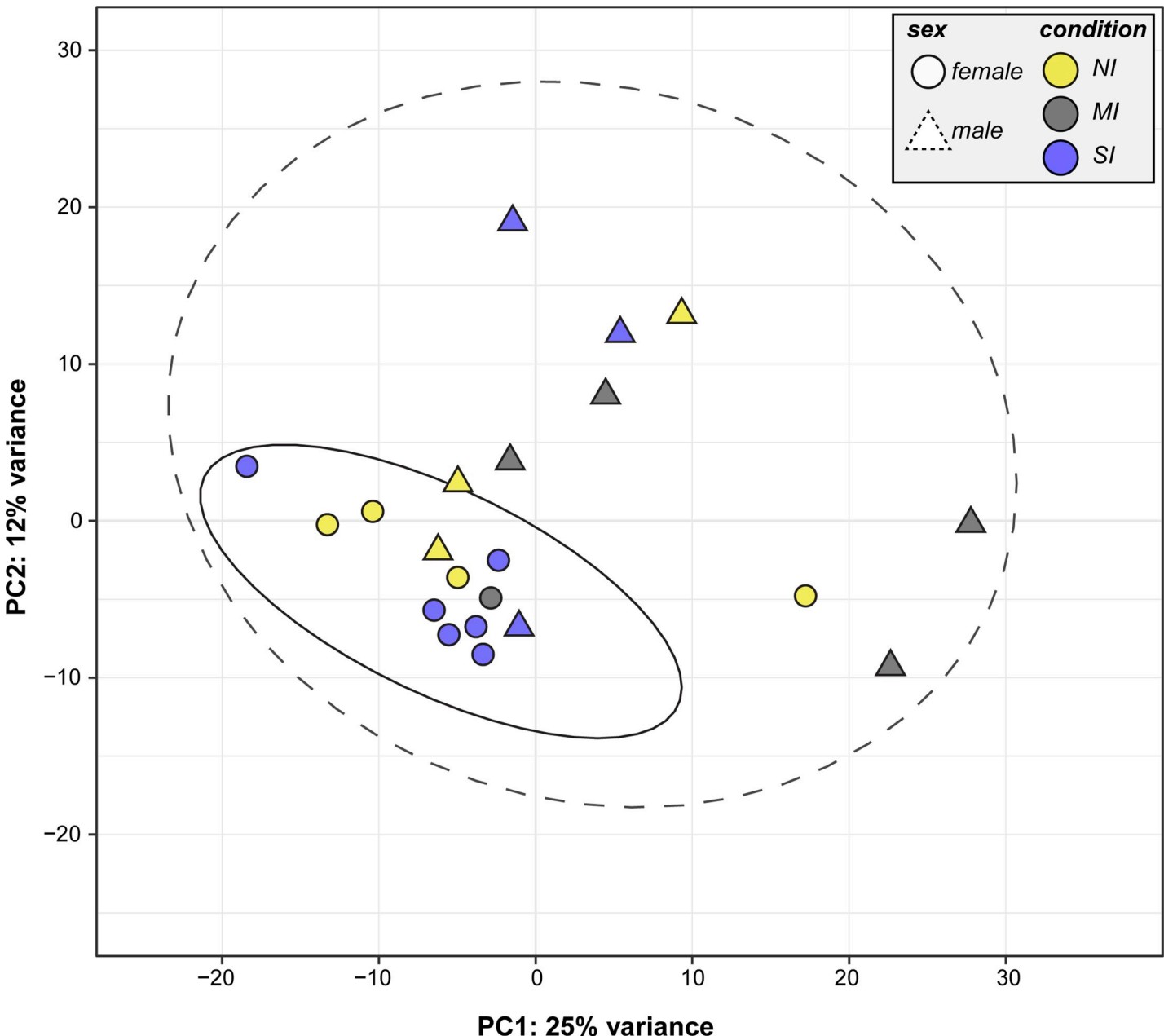

**Fig 1. Principal Components Analysis (PCA) of gene expression data explaining 37% of overall data variance.** Shapes are colored according to experimental groups (NI-No Infection, MI- Medium Infection, SI-Severe Infection). Circles represent female tortoises, squares represent male tortoises. Variables with differentially expressed genes include immune experimental group, sex, and venipuncture site (see S2 Fig).

to control (NI) with large log2-fold changes of 14.1–16.4 (SI-NI comparison values). In other words, these five genes are transcribed to differing degrees depending on whether a tortoise is infected with *Mycoplasma spp.* or not. There were 61 enriched GO terms for Biological Processes (Table 2, S3 Table), but many of these contained only one or a few DEGs. The analysis produced zero enriched KEGG pathways. Semantic clustering of enriched GO terms using REVIGO yielded eight major clusters including cysteine catabolism, response to thyroid hormone, regulation of type 1 interferon production, and regulation of fibroblast apoptotic process (Fig 3A).

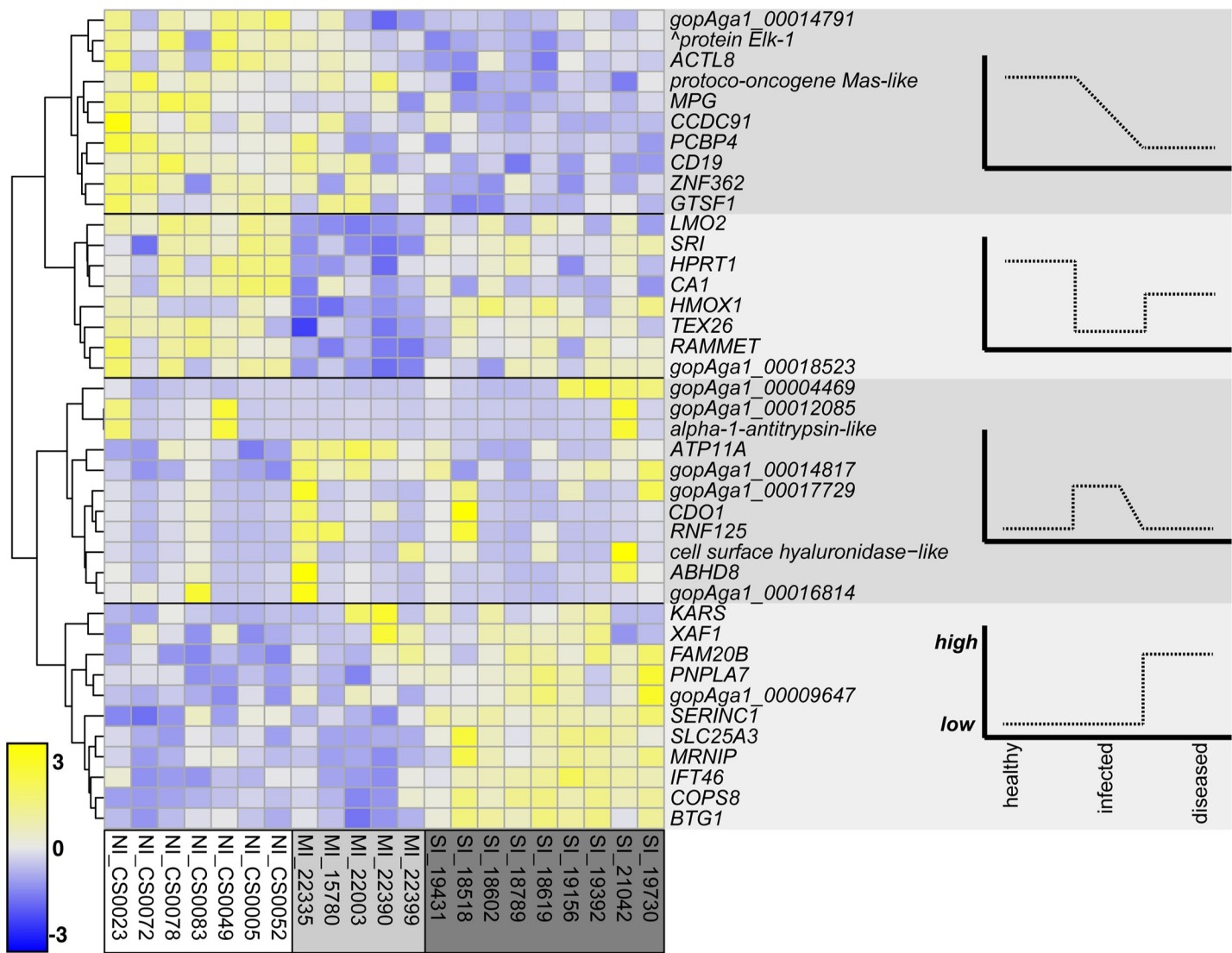

**Fig 2. Heatmap of 40 genes that are uniquely differentially expressed by experimental immune group.** Genes are clustered by the Ward method according to Manhattan distance. All genes shown (rows) are statistically significantly expressed (adjusted $\alpha < 0.05$). The tree on the left shows four clusters that reflect qualitatively different expression patterns, depicted in graphical schematics on the right-hand panel. These patterns can be further examined in future work. Color scale presents the amount of expression. Expression values are mean-centered, regularized log counts and colors are represented as z-score values. (^ ETS domain–containing protein Elk −1).

There were 1,037 genes that were uniquely differentially expressed between females and males (Fig 4, S3 Appendix). Of these, 368 were upregulated in females relative to males with the greatest log2-fold change occurring in *PACSIN3* (4.6), *SH3GL3* (4.3), gopAga1_00019967 (2.9), gopAga1_00019277 (2.1), *ZSWIM4* (2.1). There were 669 genes downregulated in females relative to males and the DEGs with the greatest log2-fold change were *RGCC* (-5.6), *SMC2* (-3.0), *CDK1* (-2.9), *NEK2* (-2.8), gopAga1_00016417 (-2.7). The g:Profiler results for sex-biased DEGs produced 116 enriched Biological Processes and 79 enriched KEGG pathways (Tables 3 and 4, S4 Table, S4 Appendix). REVIGO produced 14 semantic clusters including iron ion transport, iron ion homeostasis, response to UV-C, regulation of interferon-beta production, cellular response to organonitrogen compound, and regulation of lysosomal protein catabolism (Fig 3B).

**Table 2. Significantly enriched Gene Ontology (GO) terms for immune group analysis.**

| P value | GO ID | GO Term | No. of genes | Associated differentially expressed genes |
|---|---|---|---|---|
| 0.042 | GO:0005737 | cytoplasm | 17 | *IFT46, SRI, HMOX1, SERINC1, GTSF1, HPRT1, CCDC91, ATP11A, KARS, CDO1, RNF125, FAM20B, COPS8, BTG1, PNPLA7, XAF1, SLC25A3* |
| 0.047 | GO:0003824 | catalytic activity | 11 | *SRI, HMOX1, MPG, HPRT1, CA1, ATP11A, KARS, CDO1, RNF125, FAM20B, PNPLA7* |
| 0.041 | GO:0006807 | nitrogen compound metabolic process | 11 | *SRI,SERINC1,MPG,MRNIP,HPRT1,KARS,LMO2,CDO1,RNF125,COPS8,BTG1* |
| 0.041 | GO:0005783 | endoplasmic reticulum | 6 | *SRI,HMOX1,SERINC1,ATP11A,RNF125,PNPLA7* |
| 0.041 | GO:0010033 | response to organic substance | 6 | *SRI,HPRT1,KARS,LMO2,CDO1,RNF125* |
| 0.047 | GO:0002252 | immune effector process | 3 | *HPRT1,KARS,RNF125* |
| 0.041 | GO:0008285 | negative regulation of cell proliferation | 3 | *KARS,COPS8,BTG1* |
| 0.041 | GO:0031325 | positive regulation of cellular metabolic process | 3 | *LMO2,RNF125,COPS8* |
| 0.041 | GO:0002275 | myeloid cell activation involved in immune response | 2 | *HMOX1,KARS* |
| 0.041 | GO:0002718 | regulation of cytokine production involved in immune response | 2 | *HMOX1,KARS* |
| 0.041 | GO:0004620 | phospholipase activity | 2 | *HMOX1,PNPLA7* |
| 0.042 | GO:0009410 | response to xenobiotic stimulus | 2 | *HPRT1,CDO1* |
| 0.042 | GO:0031349 | positive regulation of defense response | 2 | *KARS,RNF125* |
| 0.042 | GO:0010212 | response to ionizing radiation | 2 | *MRNIP,KARS* |
| 0.041 | GO:0051279 | regulation of release of sequestered calcium ion into cytosol | 2 | *SRI,CD19* |

These are the significantly enriched ($\alpha \leq 0.05$) GO terms with the highest number of differentially expressed genes among immune experimental groups (Medium Infection, Severe Infection) relative to control groups (No Infection) for which gene information is available (this includes biological process, cellular component, and molecular functions).

Eight genes were differentially expressed both in the experimental immune and sex-biased analyses, which we refer to here as sex-biased immune genes (Fig 5). These sex-biased immune genes were identified as *TMEM135*, *ST6GALNAC4*, *IGHE*, *TRIM3*, and TRIM68-like, gopAga1_00007704, gopAga1_00017285, and gopAga1_00017523.

## Discussion

Infections from pathogenic bacteria such as *Mycoplasma spp.* impact the morbidity and mortality of both wild and captive tortoise populations, which we use to understand more about the general immune system of non-avian reptiles. To do this we analyzed the transcriptional response of Mojave desert tortoises to infection by *M. agassizii*. We identified 40 uniquely differentially expressed genes among experimental immune groups (Fig 2), of which 5 genes were differentially expressed both in medium infection (MI) and severe infection (SI) groups relative to control (NI) animals. Given the strong health differences among these groups, it was surprising to discover that transcription in desert tortoises was foremost influenced by sex with 1,037 genes uniquely differentially expressed between males and females (Fig 4; S3 Appendix). We identified eight genes that were significantly differentially expressed in both analyses, which we consider to be sex-biased immune genes (Fig 5). Finally, results showed that the method most accepted to draw blood from tortoises (subcarapacial venipuncture) is not ideal for gene expression analysis due to varying amounts of lymph aspirate that biases the gene expression signal based on the amount of aspirate in the sample. Our results provide a systemic view of the effects of mycoplasmosis and sex-biased transcriptional differences in desert tortoises that will aid future management and conservation practices and shed new light on the immune response of non-avian reptiles broadly.

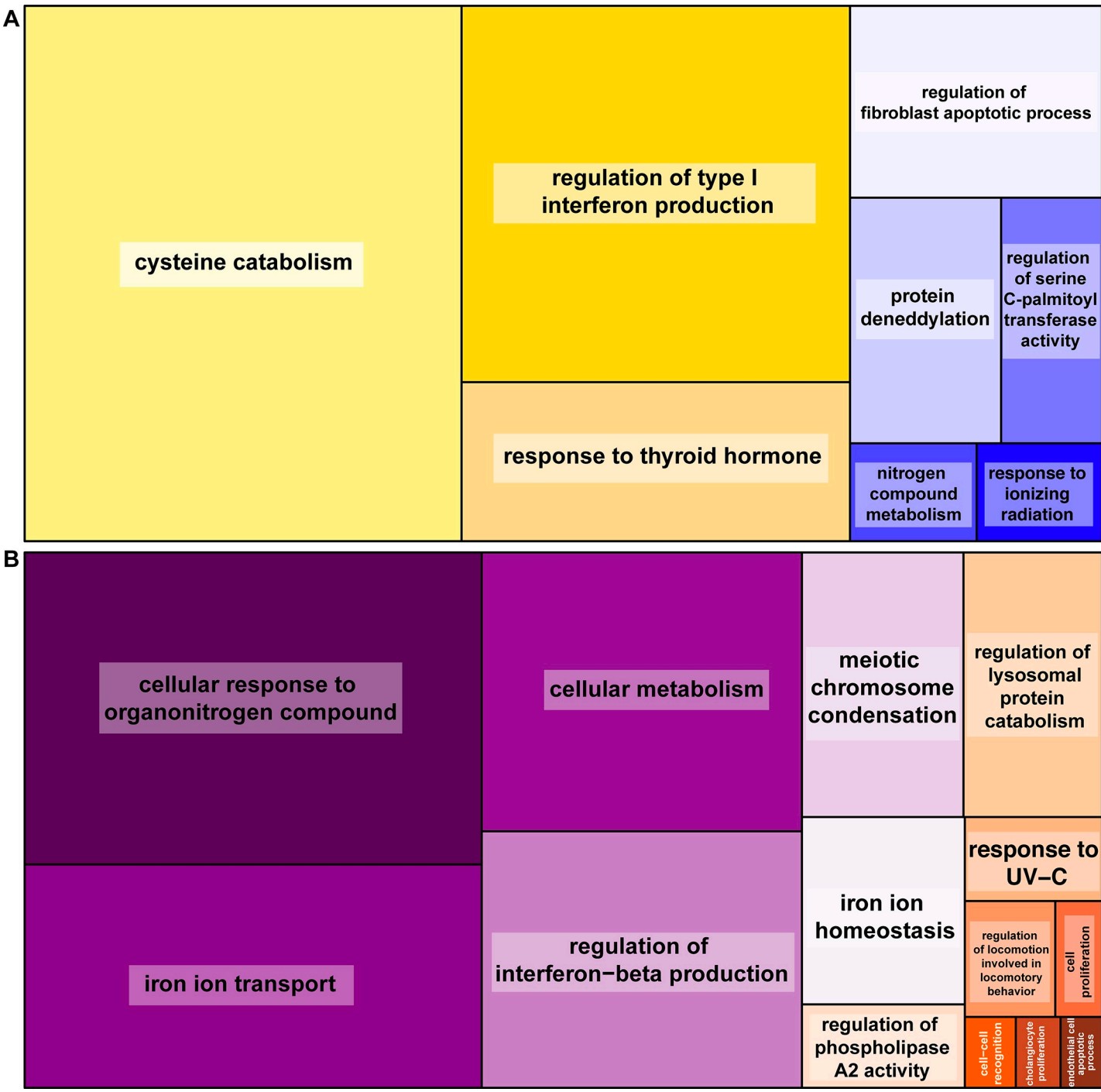

**Fig 3. REVIGO treemaps for genes differentially expressed based on experimental immune group or sex.** REVIGO treemaps showing semantically clustered enriched GO terms (colored tiles) with box size proportional to normalized adjusted p-value and clusters are shown for (A) experimental immune groups (top), and (B) sex-biased among males and females (bottom). The white labels on panel B are cell-cell recognition, cholangiocyte proliferation, and endothelial cell apoptotic process.

### Infection-based differential expression

Tortoises, like many ectotherms, rely on broad non-specific innate immune responses such as non-specific leukocytes, lysozymes, antimicrobial peptides, the complement pathway and

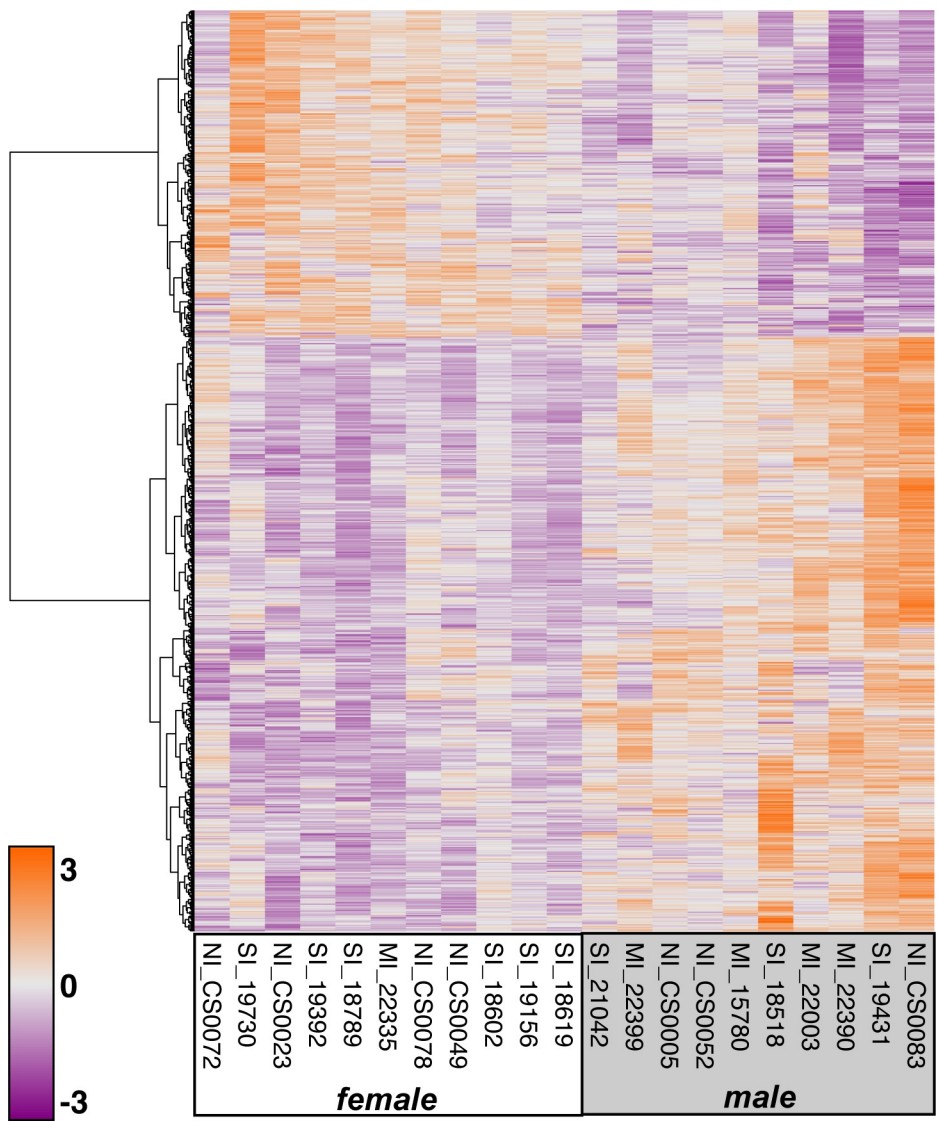

**Fig 4. Heatmap of 1,037 genes uniquely differentially expressed by sex.** Genes are clustered by the Ward method according to Manhattan distance. All genes shown (rows) are statistically significantly expressed (adjusted $\alpha < 0.05$). Color scale presents the amount of expression. Expression values are mean-centered, regularized log counts and colors are represented as z-score values.

phagocytic B cells as primary lines of defense against pathogens [1,54,55]. Adaptive immune reactions mediated by T and B cells are induced in tortoises; however, their cell-mediated and humoral responses may be slow (weeks to years; [29,31]) or fail to develop into novel antigens [56], and do not consistently demonstrate evidence of memory response [1,57]. Indeed, we found many general immune-related responses to be enriched among experimental immune groups, including GO:0002275 (myeloid cell activation involved in immune response), GO:0009410 (response to xenobiotic stimulus), and GO:0002252 (immune effector process).

Previous studies on turtles revealed expression of immune-related genes [2,31,38,58–63] associated with both innate and adaptive immune functions. We expected differential expression of cytokines (e.g. IFN-γ, interleukins, TNFα1) to be higher in tortoises with *M. agassizii* infection because they play important roles in modulating host defense responses to

**Table 3. The 25 significantly enriched (α ≤ 0.01) sex-biased Gene Ontology (GO) biological process categories.**

| Adj. p value | GO ID | GO Term | No. of genes | Associated differentially expressed genes |
|---|---|---|---|---|
| 5.29E-25 | GO:0044237 | cellular metabolic process | 433 | *TRIM25, PEMT, DAZAP2, C1D, SMARCB1, TCF25, BRPF1, DDX18, CHORDC1, RAC1, RMND5A, GTF2F1, RELB, RAB8A, DVL3, RUVBL2, DNMT1, IKBKG, CLCN3, CUL3* |
| 3.75E-16 | GO:0019222 | regulation of metabolic process | 298 | *TRIM25, PEMT, DAZAP2, C1D, SMARCB1, TCF25, BRPF1, CHORDC1, RAC1, GTF2F1, RELB, RAB8A, DVL3, RUVBL2, DNMT1, IKBKG, CLCN3, CUL3, MSH6, DNAJB1* |
| 1.27E-16 | GO:0006996 | organelle organization | 200 | *SMARCB1, SEC24B, BRPF1, CHORDC1, RAC1, ACP2, RAB8A, RUVBL2, DNMT1, CLCN3, CUL3, LETM1, MSH6, RANBP1, UQCC1, INPP5K, TEP1, KIF5B, CCT4, RALA* |
| 1.02E-16 | GO:0051641 | cellular localization | 153 | *SEC24B, RAC1, CSE1L, AP2A2, RAB8A, DVL3, RUVBL2, IKBKG, CLCN3, CUL3, LETM1, RANBP1, INPP5K, KIF5B, CCT4, PHAX, PRPF31, RALA, DERL3, UBE2G2* |
| 7.45E-6 | GO:0070887 | cellular response to chemical stimulus | 130 | *TRIM25, SMARCB1, RAC1, RAB8A, RUVBL2, DNMT1, CUL3, TBL2, TFAP4, RANBP1, SRA1, INPP5K, KIF5B, HIPK1, GABPA, DERL3, UBE2G2, RNMT, GIT1, DNAJB9, NFE2L2* |
| 2.95E-5 | GO:0035556 | intracellular signal transduction | 116 | *TRIM25, RAC1, RELB, DVL3, DNMT1, IKBKG, CUL3, MSH6, TFAP4, INPP5K, STIMATE, HIPK1, GIT1, NFE2L2, DYRK3, MAPKAPK2, BRCA1, SUZ12, ATAD5, CRLF3* |
| 3.88E-3 | GO:0008283 | cell proliferation | 95 | *PEMT, SMARCB1, BRPF1, RAC1, DNMT1, TFAP4, SRA1, HIPK1, ODC1, BRCA1, SUZ12, ATAD5, RAB5A, CDK5RAP3, MYDGF, CDK1, PSEN1, CCAR1, RPS6KB1, GGNBP2* |
| 1.53E-6 | GO:0033554 | cellular response to stress | 78 | *TRIM25, SMARCB1, CHORDC1, RELB, DVL3, CUL3, TBL2, DNAJB1, TFAP4, HIPK1, DERL3, UBE2G2, DNAJB9, NFE2L2, DYRK3, MAPKAPK2, ATAD5, CDK5RAP3, CDK1, PSEN1* |
| 7.34E-4 | GO:0071417 | cellular response to organonitrogen compound | 23 | *RAB8A, DNMT1, GABPA, PSEN1, HSP90B1, OSBPL8, RPS6KB1, SLC6A4, DMTN, RANGAP1, HSF1, PDPK1, ZEB1, PTPN1, PIK3R3, ACTB, BLM, CISH, ATP7A, LEPROT* |
| 1.25E-2 | GO:0007169 | transmembrane receptor protein tyrosine kinase signaling pathway | 19 | *GIT1, MAPKAPK2, PSEN1, OSBPL8, RPS6KB1, PDCD6, DOK2, PDPK1, RABGEF1, PTPN1, PIK3R3, SS18, LMTK2, RBPJ, PRKD2, PIK3R1, NDST1, ERBB2, MAPK1* |
| 4.35E-2 | GO:0030335 | positive regulation of cell migration | 18 | *RAC1, MIEN1, RAB5A, CCAR1, RPS6KB1, PDCD6, DMTN, NIPBL, PDPK1, ACTR3, SDCBP, PLAA, FADD, ATP7A, ITGA2B, PRKD2, PIK3R1, HMGB1* |
| 4.13E-4 | GO:0030099 | myeloid cell differentiation | 16 | *GABPA, PABPC4, DYRK3, PSEN1, PAFAH1B1, ACIN1, TFRC, SP3, FADD, CUL4A, IREB2, OSTM1, RBPJ, PIK3R1, SOX6, HMGB1* |
| 3.61E-3 | GO:0043583 | ear development | 14 | *SEC24B, RAC1, DVL3, MAPKAPK2, ABR, CEP290, PAFAH1B1, TRIP11, NIPBL, SCRIB, ZEB1, MKS1, RBPJ, MAPK1* |
| 9.29E-3 | GO:0072175 | epithelial tube formation | 11 | *SEC24B, BRPF1, DVL3, RALA, NUP50, CEP290, HNF1B, SCRIB, IFT57, MKS1, IPMK* |
| 1.58E-2 | GO:0043149 | stress fiber assembly | 9 | *RAC1, CUL3, INPP5K, RGCC, SORBS3, MKKS, WAS, ARAP1, PIK3R1* |
| 2.85E-5 | GO:0032648 | regulation of interferon-beta production | 9 | *POLR3D, RELB, YY1, IFNAR1, RIOK3, IFIH1, POLR3C, TRAF3IP1, HMGB1* |
| 3.79E-2 | GO:0007224 | smoothened signaling pathway | 9 | *HIPK1, TROVE2, ULK3, IFT57, STK36, MKS1, TRAF3IP1, NDST1, CENPJ* |
| 1.58E-2 | GO:0030038 | contractile actin filament bundle assembly | 9 | *RAC1, CUL3, INPP5K, RGCC, SORBS3, MKKS, WAS, ARAP1, PIK3R1* |
| 9.18E-4 | GO:0002066 | columnar/cuboidal epithelial cell development | 8 | *SEC24B, RAC1, PAFAH1B1, SCRIB, PDPK1, SIDT2, RFX3, YIPF6* |
| 1.21E-2 | GO:0001738 | morphogenesis of a polarized epithelium | 7 | *SEC24B, RAC1, DVL3, PAFAH1B1, MKS1, TRAF3IP1, EXOC5* |
| 8.04E-3 | GO:0060113 | inner ear receptor cell differentiation | 7 | *SEC24B, RAC1, PAFAH1B1, TRIP11, SCRIB, MKS1, RBPJ* |
| 1.02E-2 | GO:0019080 | viral gene expression | 7 | *SMARCB1, TFAP4, INPP5K, DENR, CCNT2, TARDBP, PCBP2* |
| 2.02E-2 | GO:0042771 | intrinsic apoptotic signaling pathway in response to DNA damage by p53 class mediator | 6 | *HIPK1, ATAD5, BAG6, SHISA5, TOPORS, BRCA2* |
| 4.9E-2 | GO:0072577 | endothelial cell apoptotic process | 6 | *HIPK1, NFE2L2, RGCC, PDPK1, PAK4, PRKCI* |

*(Continued)*

**Table 3.** (Continued)

| Adj. p value | GO ID | GO Term | No. of genes | Associated differentially expressed genes |
|---|---|---|---|---|
| 3.47E-2 | GO:0042149 | cellular response to glucose starvation | 5 | *TBL2, NFE2L2, HSPA5, SZT2, HIGD1A* |

The 25 significantly enriched (α ≤ 0.01) sex-biased Gene Ontology (GO) Biological Process categories with the highest number of DEGs (adjusted p values are provided, GO processes are ranked by number of genes, only the first 20 genes in each category are listed). For a complete list of DEGs and the 116 significant GO terms see S3 and S4 Appendices.

immediate and long-term pathogen exposure. While related categories GO:0002718 (regulation of cytokine production involved in immune response, Table 2) and regulation of type I interferon production (Fig 3A) were enriched, we did not find these specific genes to be differentially expressed (Table 2). Moreover, western painted turtles (*Chrysemys picta bellii*; [58]) demonstrated a unique repertoire of toll-like receptors (TLRs) including TLR15-like receptor known in the response of birds to bacterial pathogens [64]. In our data we found related pattern recognition signaling categories such as GO:0039529 and GO:0039536 (RIG-I signaling pathway) as well as GO:1900745 (regulation of p38 MAPK cascade) and GO:1901224 (regulation of NIK/ NF-κB signaling), which are involved in innate and adaptive immune activation and maintenance. On the other hand, these pathways are important mediators of inflammation and if prolonged, may exacerbate the effects of *M. agassizii* infection and URTD.

Chinese soft-shelled turtles (*Pelodiscus sinensis*) infected with pathogenic bacteria also demonstrated differential expression of innate immune genes including IL-8, serum amyloid A (SAA), *CD9*, *CD59*, activating transcription factor 4 (*ATF4*) and cathepsin L genes [62], pointing to an initial non-specific innate immune response, followed by later moderate adaptive responses to combat remaining pathogens. Given that this study was designed to assess chronic rather than acute immunological responses, it was not surprising that these five genes differentially expressed among the immune groups. However, we did observe that *CD19* was downregulated in both the desert tortoise MI and SI groups relative to uninfected controls, although not significantly in the MI group. CD19 is an antigen expressed by both subsets of B lymphocytes. B1 cells are innate-like effectors that produce natural antibodies and exhibit phagocytic activity in fish, amphibians, and reptiles, including turtles/tortoises [1,65,66]. Additionally, B2 cells are responsible for generating antigen-specific antibodies against foreign pathogens. In Mojave desert tortoises, the mean infection intensity of *M. agassizii* is negatively correlated with the mean number of lymphocytes [65], which could provide protection via phagocytosis during early infection or long-term humoral immunity. Given that MI and SI individuals have been chronically infected with *M. agassizii* and tested positively for acquired antibodies, the latter is more likely in this study. Unlike the MI group, *CD19* expression was significantly reduced in SI individuals, suggesting that the suppression of B lymphocytes may increase susceptibility to infection and morbidity. Taken together, *CD19* may be a key B lymphocyte antigen in the chelonian immune response to infection, and may be a good gene target for future studies.

Of the 40 genes that were uniquely expressed among immune groups, 28 genes were previously annotated and are associated with a number of immune and metabolic processes. Genes *ABHD8*, *CDO1*, *RNF125*, cell surface hyaluronidase-like, and gopAga1_00017729 were significantly upregulated in MI and SI animals with high log2-fold change values (Fig 2). Most of the differentially expressed immune genes are involved in protein production, folding, and secretory domains, with additional broad functions related to both host defenses and mitigation of host-induced inflammatory responses. For example, cysteine dioxygenase type 1 (*CDO1*) is a

**Table 4. The 25 significantly enriched (α ≤ 0.05) sex-biased KEGG pathways.**

| Adj p value | KEGG ID | KEGG name | No. of genes | Differentially expressed genes |
|---|---|---|---|---|
| 5.41E-3 | KEGG:05165 | Human papillomavirus infection | 29 | DVL3, IKBKG, HDAC5, PPP2R5C, HDAC2, PSEN1, PPP2CA, RPS6KB1, PPP2R5E, SCRIB, IFNAR1, HDAC3, TCF7L2, UBE3A, LAMC1, RBL1, PIK3R3, FADD, ATP6V1H, ITGA2B |
| 5.41E-3 | KEGG:04144 | Endocytosis | 23 | AP2A2, RAB8A, KIF5B, GIT1, CYTH1, RAB5A, VPS26A, RUFY1, TFRC, USP8, CHMP5, SH3GL3, WAS, VPS35, ARAP1, SNX2, PRKCI, WIPF2, CLTC, ARF1 |
| 3.17E-3 | KEGG:04141 | Protein processing in endoplasmic reticulum | 18 | SEC24B, DNAJB1, DERL3, UBE2G2, NFE2L2, HSP90B1, HSP90AA1, SSR1, DNAJC3, PDIA4, HSPA5, SEC62, PLAA, EIF2AK1, EIF2AK3, DNAJA2, SEC24A, NPLOC4 |
| 5.41E-3 | KEGG:04530 | Tight junction | 16 | MARVELD3, RAC1, RAB8A, ACTR2, PPP2CA, HSPA4, SCRIB, ACTR3, ACTB, WAS, MYL12B, RAPGEF6, PRKCI, PPP2R2D, PATJ, ERBB2 |
| 2.60E-2 | KEGG:04151 | PI3K-Akt signaling pathway | 16 | RAC1, PPP2R5C, YWHAB, YWHAH, HSP90B1, PPP2CA, RPS6KB1, HSP90AA1, PDPK1, EIF4E, PIK3R3, ITGA2B, PIK3R1, PPP2R2D, ERBB2, MAPK1 |
| 3.17E-3 | KEGG:05164 | Influenza A | 15 | TRIM25, DNAJB1, KPNA2, XPO1, IFNAR1, IFIH1, PIK3R3, ACTB, EIF2AK3, OAS3, HNRNPUL1, PIK3R1, NXT2, MAPK1, CYCS |
| 1.95E-2 | KEGG:05168 | Herpes simplex infection | 14 | PPP1CB, CDK1, HCFC2, CDC34, USP7, IFNAR1, CSNK2B, TAF5, IFIH1, FADD, EIF2AK3, OAS3, CYCS, CSNK2A1 |
| 2.11E-2 | KEGG:03013 | RNA transport | 14 | PHAX, PABPC4, NUP50, XPO1, ACIN1, RANGAP1, DDX20, EIF4E, NCBP1, UPF3B, NXT2, THOC7, EIF2S2, EIF4EBP3 |
| 4.19E-2 | KEGG:05225 | Hepatocellular carcinoma | 14 | SMARCB1, DVL3, NFE2L2, RPS6KB1, SMARCD1, TCF7L2, SMARCC2, ACTL6A, PIK3R3, ACTB, PIK3R1, MAPK1, SMARCD2, RPS6KB2 |
| 1.16E-2 | KEGG:03040 | Spliceosome | 14 | PUF60, PRPF31, EFTUD2, DDX46, ACIN1, PLRG1, NCBP1, SRSF10, SYF2, SRSF9, USP39, PPIH, PRPF4, SF3B4 |
| 8.13E-3 | KEGG:04120 | Ubiquitin mediated proteolysis | 14 | CUL3, UBE2G2, BRCA1, HERC4, FBXW11, CDC34, HUWE1, UBE3A, CUL4A, BIRC3, CUL5, PIAS1, UBA6, UBA2 |
| 5.41E-3 | KEGG:04140 | Autophagy—animal | 13 | PPP2CA, RPS6KB1, PDPK1, MLST8, RRAGD, PIK3R3, EIF2AK3, ATG2B, PIK3R1, ATG16L2, MAPK1, HMGB1, NRBF2 |
| 1.38E-3 | KEGG:05169 | Epstein-Barr virus infection | 13 | POLR3D, YWHAB, YWHAH, HDAC2, CDK1, PSMC6, USP7, PSMD14, POLR3C, PIK3R3, RBPJ, PIK3R1, CSNK2A1 |
| 1.95E-2 | KEGG:04510 | Focal adhesion | 12 | RAC1, PPP1CB, PDPK1, FLNB, PIK3R3, ACTB, BIRC3, ITGA2B, MYL12B, PIK3R1, ERBB2, MAPK1 |
| 2.03E-2 | KEGG:05160 | Hepatitis C | 12 | IKBKG, PPP2CA, IFNAR1, PDPK1, PIK3R3, EIF2AK1, EIF2AK3, PIAS1, OAS3, PIK3R1, PPP2R2D, MAPK1 |
| 5.41E-3 | KEGG:03015 | mRNA surveillance pathway | 12 | RNMT, PABPC4, PPP1CB, PPP2R5C, PPP2CA, PPP2R5E, ACIN1, ETF1, NCBP1, UPF3B, PPP2R2D, NXT2 |
| 2.63E-2 | KEGG:04152 | AMPK signaling pathway | 11 | RAB8A, PPP2R5C, PPP2CA, RPS6KB1, PPP2R5E, HMGCR, PDPK1, PIK3R3, PIK3R1, PPP2R2D, RPS6KB2 |
| 2.03E-2 | KEGG:04390 | Hippo signaling pathway | 11 | DVL3, PPP1CB, YWHAB, YWHAH, FBXW11, PPP2CA, SCRIB, ACTB, PPP2R2D, MOB1A, PATJ |
| 1.38E-3 | KEGG:05203 | Viral carcinogenesis | 11 | GTF2H1, MAPKAPK2, YWHAB, CDK1, GTF2A1, USP7, UBE3A, PIK3R3, RBPJ, PIK3R1, MAPK1 |
| 8.13E-3 | KEGG:05212 | Pancreatic cancer | 10 | RAC1, IKBKG, RALA, RPS6KB1, PIK3R3, BRCA2, PIK3R1, ERBB2, MAPK1, RPS6KB2 |
| 1.38E-3 | KEGG:04114 | Oocyte meiosis | 10 | PPP1CB, PPP2R5C, YWHAB, YWHAH, CDK1, PPP3CB, SLK, MAD2L1, PPP3R1, MAPK1 |
| 8.02E-3 | KEGG:04910 | Insulin signaling pathway | 10 | PPP1CB, RPS6KB1, PRKAR1A, PDPK1, PTPN1, EIF4E, PIK3R3, PKLR, PIK3R1, MAPK1 |
| 1.56E-2 | KEGG:05210 | Colorectal cancer | 10 | RAC1, MSH6, RALA, RPS6KB1, TCF7L2, PIK3R3, PIK3R1, MAPK1, CYCS, RPS6KB2 |
| 1.12E-2 | KEGG:03008 | Ribosome biogenesis in eukaryotes | 9 | XPO1, CSNK2B, RBM28, MPHOSPH10, XRN1, TBL3, NXT2, CSNK2A1, RIOK2 |
| 8.13E-3 | KEGG:05170 | Human immunodeficiency virus 1 infection | 8 | CDK1, RPS6KB1, PIK3R3, FADD, CUL4A, PPP3R1, MAPK1, CYCS |

The 25 significantly enriched (α ≤ 0.05) sex-biased KEGG pathways (Kyoto Encyclopedia of Genes and Genomes) that have the highest number of differentially expressed genes (significance based on adjusted p values). For the complete list of 79 pathways see S4 Table.

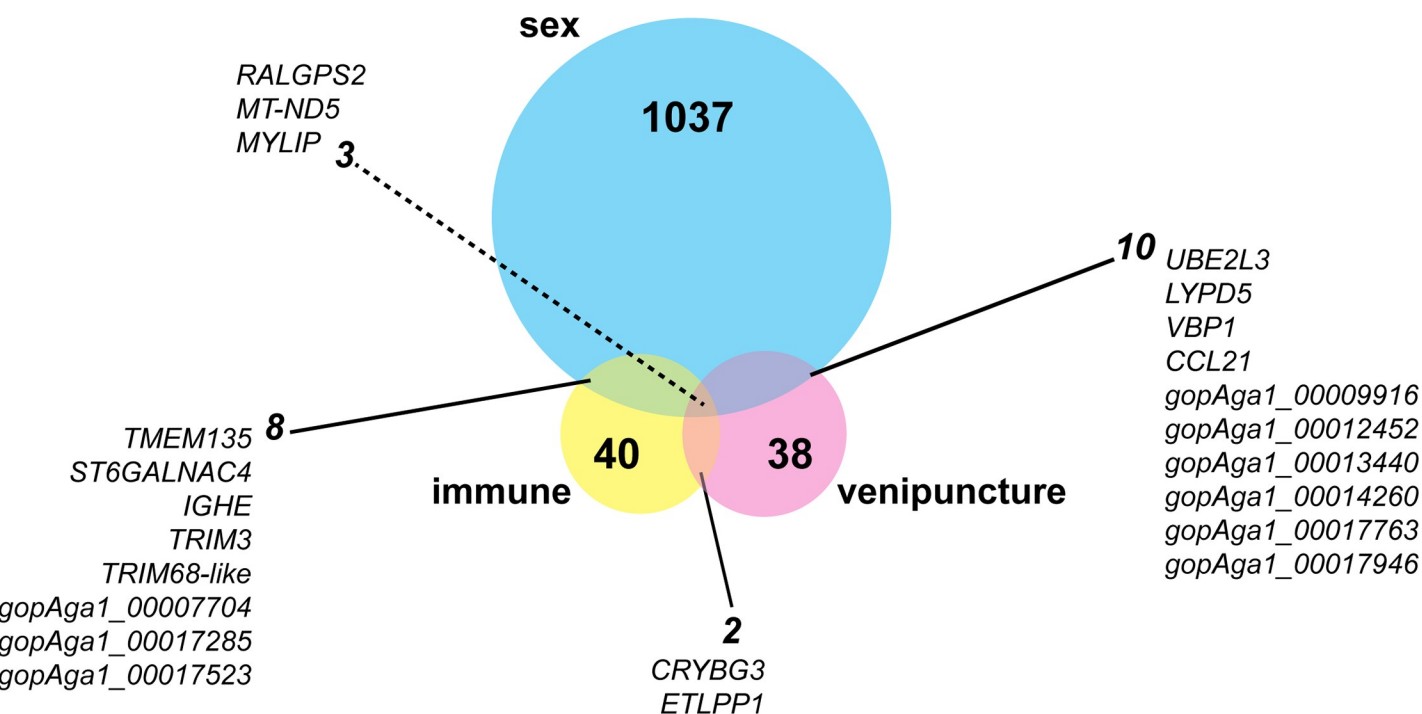

**Fig 5. Venn diagram of shared differentially expressed genes among sex, experimental immune group, and venipuncture site.** Dotted line points to the three genes differentially expressed in all comparisons.

protein-coding gene that responds to glucagon, xenobiotic stimuli, bacterial insults, and organic substances, leading to enzymatic cysteine catabolic processes and dioxygenase activity (Table 2; [67]). These processes aid in providing essential biomolecules such as vitamins, cofactors, antioxidants, and many defense compounds that frontline innate immune cells (e.g. macrophages and dendritic cells) need to protect cells or neutralize targeted antigens (e.g., invading bacteria). Relative to the control group, *RNF125* was highly upregulated in both MI and SI groups with respective log2-fold changes of 14.9 and 16.5. As a ubiquitin ligase, RNF125 has been shown to reduce inflammatory signaling by targeting RIG-I [68] as well as regulating viral transcription in peripheral blood mononuclear cells [69]. *RNF125* is primarily expressed in lymphoid tissues and is a positive regulator of T lymphocyte activation [70–72], which may be critical for modulating inflammatory pathways and adaptive host defense against infection in desert tortoises.

Pathways with enriched DEGs included those involved in immune host defenses and regulatory processes. For example, cysteine catabolism plays an important role in many proteins and immune defenses such as mRNA transcription, regulation of macrophage chemotaxis, heme oxidation, etc. (Fig 3A; [73]). Other pathways such as regulation of type I interferon production (e.g. myeloid cell activation, basophil activation, immune effector processes) include a family of cytokines that are critically important in controlling host innate and adaptive immune responses to viral and bacterial infections, and other inflammatory responses (Fig 3A; [74]). There were also differential processes associated with regulation of thyroid hormones [75] and ionizing radiation [76] conditions. These enriched processes associate with modulating immune activities such as chemotaxis, phagocytosis, generation of reactive oxygen species (ROS), and cytokine synthesis at the cellular level based on hypo- and hyperthyroid conditions. Alternatively, they can interfere with the interactions of targeted cells such as dendritic cells and lymphocytes between innate and adaptive cell-mediated immunity, respectively [75].

## Sex-biased differential expression

Sex is increasingly recognized as an important effector of the immune response [77–79]. Indeed, our results show striking differences in transcription between male and female tortoises evidenced by 1,037 sex-biased DEGs that yielded 116 enriched Biological Processes and 79 enriched KEGG pathways (Figs 3B and 4; Tables 3 and 4). Previous literature has shown that exposure to pathogens elicits sex-biased differential immunological responses. Females generally initiate stronger innate and adaptive immune responses, which can promote faster clearance of pathogens. This has been observed both in birds where females show increased T lymphocyte proliferation after parasite exposure compared to males [80] as well as in lizards where female-derived macrophages demonstrated greater phagocytic activity than male-derived macrophages [81]. In humans, stimulation of TLR7 in plasmacytoid dendritic cells induces significantly greater expression of IFNα in females [82], and females show enhanced antibody responses with higher B cell numbers [83]. Consistent with the literature, our findings show that females exhibit upregulation of genes associated with cytokine production and host defense compared to males, including *POLR3D*, *POLR3C*, *MAPKAPK2*, *MR1*, *ARID5A*, *SARS*, *SETD6*, and *PCBP2* (Table 3, S1 Appendix).

While females generally elicit stronger immune responses, this pattern is sometimes reversed. In this study, male tortoises displayed increased expression for biological processes involved in myeloid cell differentiation and activation, leukocyte mediated immunity, and positive regulation of tumor necrosis factor production (Figs 3B and 4; Tables 3 and 4). In mice, macrophages from males produced greater proinflammatory cytokines during the acute phase than their female counterparts [84]. Similarly, whole blood and neutrophils from human males produced greater levels of tumor necrosis factor than females [85,86]. Dysregulation of the immune response can lead to chronic infection or sepsis, which males are more prone to develop than females [87,88]. In contrast, females are more susceptible to inflammatory and autoimmune diseases as a result of stronger mounted immune responses [87]. Indeed, the *TNIP1* gene is associated clinically with female-biased autoimmune diseases such as systemic lupus erythematosus and systemic sclerosis [88–90], which are diseases characteristic of an over-active immune system and here we also find *TNIP1* to be significantly upregulated in female tortoises. While immune cell levels are not significantly different between male and female desert tortoises with no history of infection [12], our findings indicate that there is a sex-biased immune response following pathogen exposure. Whether immune cells vary between sex following infection is unknown but should be tested in future studies. Additionally, because immune cell function and levels change with seasonality and temperature in tortoises [12,106], it is possible that sex-biased gene expression is also affected by these factors. Overall, the sex-biased patterns in the literature are complex and sex-biased outcomes of *Mycoplasma* spp. infection and URTD warrants further study.

Sex-biased immune responses depend on genetics and hormones. However, turtles (including the desert tortoise) lack sex chromosomes and instead have temperature-based sex determination [14]. How this fact affects immune gene expression, if at all, is unknown. Hormones, on the other hand, fluctuate seasonally and are responsive to changes in temperature in non-avian reptiles. Additionally, many sex hormone receptors directly regulate gene transcription by translocating to the nucleus and binding hormone response elements when activated. This suggests that hormones likely play a key role in sex-biased gene expression and immunity.

Indeed, female anole lizards suppress female-biased gene expression and exhibit higher levels of male biased genes when treated with testosterone [91]. Testosterone, which is higher in male than female vertebrates regardless of the mode of sex determination, also has immunosuppressive effects on immune cell activity [92]. Testosterone reduces the production of pro-

inflammatory cytokines in mammals [93] and decreases cell-mediated immune reactions in birds, with greater immune suppression occurring in males than females [94]. In this study, the *TMF1* gene, which is associated with the GO category for testosterone secretion, is significantly upregulated in male tortoises as expected. Whether testosterone causes immunosuppression in tortoises has not been investigated, but it is worth noting that testosterone levels in the desert tortoise vary seasonally [95]. This means that if there is an immunosuppressive effect of testosterone in tortoises then that immunosuppression may also vary seasonally. Testosterone levels begin to rise in April through July and peak in late August and September when male-male aggression, mating activity, and metabolism are greatest [96,97].

Additionally, testosterone is highly correlated with production of corticosterone [96], which also has immunosuppressive effects and has been shown to be higher on average in male tortoises relative to females [96,98]. Corticosteroids are generated as a stress response and we identified several genes upregulated in males that enriched for cellular response to stress (GO:0033554). These results together raise the question of whether immunosuppression in males during mating season predisposes them to *Mycoplasma* spp. infection due to a dampened immune response. Moreover, male tortoises also contact both sexes more frequently relative to females [30] and males travel greater distances, suggesting they may be more likely to spread *M. agassizii* (or other infections) among populations. Further investigation would be instructive for future wildlife management practices.

As expected, differentially expressed genes between male and female tortoises were enriched for metabolism, particularly those related to iron. Iron is an essential cofactor for many metabolic processes including cellular respiration, oxygen transport, and DNA synthesis. In this study, genes associated with iron homeostasis and iron ion import were expressed at higher levels in male tortoises (e.g., *TFRC*, *SLC25A37*). Transferrin receptor (*TFRC*) is required for the cellular uptake of iron through endocytosis and *SLC25A37* is involved in transporting cytosolic iron into mitochondria via Mitoferrin-1. Sex differences in iron uptake may be associated with elevated energy demands in males because male tortoises have larger home range sizes and travel larger distances relative to females [99–103], and may also expend more acute energy requirements by engaging in combative activity over mates. Perhaps this is also why males exhibited higher gene expression variability than females (Fig 1). Additionally, female desert tortoises lay their eggs in April–mid July [104,105], which requires a large investment of energy and resources. Allocation of energetic resources during this period may deplete iron levels and could further contribute to transcriptional differences related to iron homeostasis between males and females.

Cellular sequestration of iron is a recognized immunological response. Tortoises challenged with lipopolysaccharide demonstrated a reduction in plasma iron concentration [106]. Host import of iron prevents the pathogen from acquiring it, thereby limiting the rate of pathogen proliferation [107]. Given the diverse roles of iron in metabolism and immunity, future studies will be important to determine the functional effects of iron homeostasis. Fortunately, iron can be assayed easily and cheaply for wild and captive animals, lending itself as a good topic for future studies.

## Conclusions and implications for future studies

We carried out RNA-Seq and differential expression analysis to identify the systemic host response of the Mojave desert tortoise to three levels of *Mycoplasma agassizii* infection. We identified 40 uniquely differentially expressed genes associated with infection. Among these were genes involved in protein production, secretory domains, host defenses, and mitigation of host-induced inflammatory responses. The genes *ABHD8*, *CDO1*, *RNF125*, cell surface

hyaluronidase-like, and gopAga1_00017729 were upregulated in medium and severely infected animals relative to control, indicating these may be biomarkers of infection. A stronger result from these data is that sex plays a dominant role in determining gene expression, even among healthy and severely sick animals (1,037 sex-biased vs. 40 immune-based DEGs, respectively). We identified eight genes that were differentially expressed both by sex and infection status (i.e. sex-biased immune genes). Further assessment of gene expression before and during disease progression would be instructive. Because tortoises have temperature-based sex determination, effects of sex hormones are of primary interest, including the potential immunosuppressive effect of testosterone and corticosterone during select seasonal periods of their activity. For future studies, subcarapacial venipuncture may result in aspirated lymph fluid, which will confound gene expression analysis, so jugular venipuncture would be a preferred method of blood collection for gene expression studies.

## Supporting information

**S1 Table. Sequencing and mapping statistics for the 25 samples included in this study.**
*Denotes samples removed from analysis due to low sequencing depth.
(DOCX)

**S2 Table. Enriched Gene Ontology (GO) terms for biological processes that are uniquely differentially expressed based on venipuncture site.**
(DOCX)

**S3 Table. Enriched Gene Ontology (GO) terms for biological processes that are uniquely differentially expressed among experimental groups (No Infection, NI; Medium Infection, MI; Severe Infection, SI).**
(DOCX)

**S4 Table. Enriched KEGG pathways for genes uniquely differentially expressed based on sex.**
(DOCX)

**S1 Fig. Principal Components Analysis (PCA) of gene expression data based on different variables.** Color-coded by additional variables relevant to the experimental design: (A) low coverage samples, (B) venipuncture site, (C) wild vs. captive, (D) collection year. Only high coverage samples are shown for venipuncture site, wild vs. captive, and collection year. Venipuncture site showed a pattern and was analyzed through the DESeq2 pipeline (see S2 and S3 Figs, and S1 Appendix).
(TIF)

**S2 Fig. Heatmap of 38 genes that are uniquely differentially expressed by venipuncture site.** Subcarapacial blood draws may have aspirated lymph fluid due to proximal lymphatic vessels, however this is the preferred phlebotomy technique by management agencies. Expression values are mean-centered, regularized log counts and colors are represented as z-score values. Jugular, light blue; subcarapacial, tan.
(TIF)

**S3 Fig. REVIGO treemap for genes differentially expressed based on venipuncture site.**
(TIF)

**S1 Appendix. Differentially Expressed Genes (DEGs) by venipuncture site (all, including non-unique genes).**
(XLSX)

**S2 Appendix. Differentially Expressed Genes (DEGs) by experimental immune group (all, including non-unique genes).**
(XLSX)

**S3 Appendix. Differentially Expressed Genes (DEGs) by sex (all, including non-unique genes).**
(XLSX)

**S4 Appendix. Enriched Gene Ontology (GO) terms for biological processes that are uniquely differentially expressed based on sex.**
(XLSX)

## Acknowledgments

We thank many people for their support with this research including Christina Aiello, Josephine Braun, Patrick Emblidge, Rachel Foster, Peter Hudson, Sydney Kelly, Kenneth Nussear, and Margarete Walden.

## Author Contributions

**Conceptualization:** Cindy Xu, Greer A. Dolby, K. Kristina Drake, Todd C. Esque, Kenro Kusumi.

**Data curation:** Cindy Xu, Greer A. Dolby.

**Formal analysis:** Cindy Xu, Greer A. Dolby.

**Funding acquisition:** Greer A. Dolby, Kenro Kusumi.

**Investigation:** Cindy Xu, Greer A. Dolby, K. Kristina Drake, Todd C. Esque, Kenro Kusumi.

**Methodology:** Cindy Xu, Greer A. Dolby.

**Project administration:** Cindy Xu, Greer A. Dolby, K. Kristina Drake, Todd C. Esque, Kenro Kusumi.

**Resources:** K. Kristina Drake.

**Supervision:** Greer A. Dolby, Todd C. Esque, Kenro Kusumi.

**Validation:** Cindy Xu, Greer A. Dolby.

**Visualization:** Cindy Xu, Greer A. Dolby.

**Writing – original draft:** Cindy Xu, Greer A. Dolby, K. Kristina Drake, Kenro Kusumi.

**Writing – review & editing:** Cindy Xu, Greer A. Dolby, K. Kristina Drake, Todd C. Esque, Kenro Kusumi.

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
