## [Decision Letter · Decision Letter 0]

14 May 2020

PONE-D-20-12290

Immune and Sex-Biased Gene Expression in the threatened Mojave desert tortoise, Gopherus agassizii

PLOS ONE

Dear Dr. Kusumi,

Thank you for submitting your manuscript to PLOS ONE. After careful consideration, we feel that it has merit but does not fully meet PLOS ONE’s publication criteria as it currently stands. Therefore, we invite you to submit a revised version of the manuscript that addresses the points raised during the review process.

We would appreciate receiving your revised manuscript by Jun 28 2020 11:59PM. To enhance the reproducibility of your results, we recommend that if applicable you deposit your laboratory protocols in protocols.io, where a protocol can be assigned its own identifier (DOI) such that it can be cited independently in the future. For instructions see: http://journals.plos.org/plosone/s/submission-guidelines#loc-laboratory-protocols

We look forward to receiving your revised manuscript.

Kind regards,

Y-h. Taguchi, Dr. Sci.

Academic Editor

PLOS ONE

Journal Requirements:

2. To comply with PLOS ONE submissions requirements, please provide methods of sacrifice in the Methods section of your manuscript.

3. In your Methods section, please provide additional location information of the study sites, including geographic coordinates for the data set if available.

4. We note that you are reporting an analysis of a microarray, next-generation sequencing, or deep sequencing data set. PLOS requires that authors comply with field-specific standards for preparation, recording, and deposition of data in repositories appropriate to their field. Please upload these data to a stable, public repository (such as ArrayExpress, Gene Expression Omnibus (GEO), DNA Data Bank of Japan (DDBJ), NCBI GenBank, NCBI Sequence Read Archive, or EMBL Nucleotide Sequence Database (ENA)). In your revised cover letter, please provide the relevant accession numbers that may be used to access these data. For a full list of recommended repositories, see http://journals.plos.org/plosone/s/data-availability#loc-omics or http://journals.plos.org/plosone/s/data-availability#loc-sequencing.

'I have read the journal's policy and the authors of this manuscript have the following

competing interests: G.A.D. is a member of the Board of Directors for the Desert

Tortoise Council, a nonprofit conservation or- ganization that protects desert tortoises

and their habitats. The DTC had no involvement in the design, implementation, or

interpretation of this study.'

Additional Editor Comments (if provided):

Although the decision is major, the reviewers are quite position to this research. Please address these concerns as much as possible, especially concerns raised by the reviewer 2.

Reviewers' comments:

Reviewer's Responses to Questions

**Comments to the Author**

1. Is the manuscript technically sound, and do the data support the conclusions?

Reviewer #1: Yes

Reviewer #2: Partly

2. Has the statistical analysis been performed appropriately and rigorously? 

Reviewer #1: Yes

Reviewer #2: Yes

3. Have the authors made all data underlying the findings in their manuscript fully available?

Reviewer #1: Yes

Reviewer #2: Yes

4. Is the manuscript presented in an intelligible fashion and written in standard English?

Reviewer #1: Yes

Reviewer #2: Yes

5. Review Comments to the Author

Reviewer #1: In this study, Xu and colleagues compared blood-cell transcriptional profiles between healthy and diseased desert tortoises infected with a species-specific Mycoplasma. Besides comparing expression between treatment groups, they focused on sex-specific differences with the aim of distinguishing sex-biased immunological responses. They found that the majority of genes show sex-biased differences in expression but not to different levels of infection, for which they found a few dozen genes or less. I think the results presented here are an important contribution to different fields (i.e. genomics and evolutionary biology) and not just desert tortoise biology and conservation. The manuscript is very well written from the Methods section onwards, with the Abstract and Intro falling short, and, to my opinion, needing a bit more work. Consequentially, the two main comments that I have are:

1) The Introduction is a bit short and has room for improvement. It can also benefit from having better transitions between topics. As you said, there aren't many studies about immunity in non-avian reptiles so it would be best to have a solid background on the system. There are parts in the Discussion that I feel need to be stated in the Intro as well, such as, expectations about differences between sexes and about expression in blood-cells and how it relates to immunity. Which genes are expected to be upregulated and why?

2) As I explain below, I don't find it surprising that you are finding a strong sex-biased signal. There has been a lot of work, both experimental and theoretical, explaining sex-biased differences in gene expression, and they obviously point to sexual selection and sexual conflict. Therefore, it would be great if more of this type of literature is mentioned and discussed in the manuscript. I think you do a good job at briefly mentioning this in the discussion, but there is much more to be said, and there is nothing in the intro about this.

Other than that, I think this is a very well-done study and I'm looking forward to reading revised versions of it. I hope that you find my comments below useful.

With kind regards,

Santiago Sanchez-Ramirez

Full review:

Title

This may be journal-dependent, but the title has some words capitalized and others don't. To make it consistent you need to capitalize "threatened", "desert", and "tortoise", unless you have a good reason not to.

The title also seems a bit broad and does not highlight your main results. You could try something along the lines: "Sex-biased expression overshadows transcriptomic immune responses to infection in the Mojave Desert tortoise".

Abstract

Line 26: Why is this surprising? Differential expression between males and females in sexually reproducing species is absolutely widespread. There is a ton of evidence of sex-biased expression and theory supporting why this happens (i.e. sexual selection and sexual conflict). Sexually dimorphic phenotypes have been shaped by sexual selection continuously over millions of years, whereas immune responses to pathogens probably have more ephemeral and dynamic footprints on the genome. Here are some reviews on the subject [of sex-biased expression]:

Rice WR (1984) Sex chromosomes and the evolution of sexual dimorphism. Evolution, 38, 735–742.

Ellegren H, Parsch J (2007) The evolution of sex-biased genes and sex-biased gene expression. Nature Reviews Genetics, 8, 689–698.

Mank JE (2017) The transcriptional architecture of phenotypic dimorphism. Nature Ecology & Evolution, 1, 1–7.

Kasimatis KR, Nelson TC, Phillips PC (2017) Genomic Signatures of Sexual Conflict. Journal of Heredity, 108, 780–790.

Rowe L, Chenoweth SF, Agrawal AF (2018) The genomics of sexual conflict. The American Naturalist, 192, 274–286.

Line 31: "further investigation".

Lines 29:32: This statement is a bit long and convoluted. Please restructure. Perhaps into 2 sentences? It is also poorly connected to the previous sentence. Why "specifically"?

Line 33: "enriched GO [biological?] processes for".

Line 35: Are these all sex-biased genes or just the 8 that have sex-by-immune responses? Please clarify.

Lines 37:39: Instead of concluding with methodological suggestion, why don't you make a general statement about an most important take-home message of your results or why your findings are important for the scientific community?

Introduction

Line 42: "particularly because of how".

Lines 43-54: The order of importance of these challenges is non-trivial. Availability of genomic resources seems to me to be less imsportant than "fewer functional studies about the immunological mechanisms and pathways" for understanding innate and adaptive immune systems of non-avian reptiles. I think it makes more sense to place the genomic challenge at the end. That way you can connect the fact that more data is underway with your next paragraph on RNA-seq.

Line 55: Also check out this review paper in Nature Rev. Genetics. It quite relevant for your study:

Klein SL, Flanagan KL (2016) Sex differences in immune responses. Nature Reviews Genetics, 1–13.

Line 55: "how sex-biased differences manifest in the immune function". This is oddly written. Sex differences may influence immune function, but it is strange to think about sex differences manifesting in the immune function.

Line 58: ", such as RNA-seq, ".

Line 59: I don't think its the right place to introduce the system. Perhaps in a different paragraph. Why don't you move the first sentence in this paragraph to the last of the previous paragraph?

Lines 57-65: This paragraph is kind of weakly constructed. Why don't you introduce the system by mentioning aspects about the biology and conservation status. Then mention that Mycoplasma research is important for conservation efforts and that there has been a lot of interest to understand host-pathogen relationships.

Are there any kind of expectations about differentially expressed genes under these experimental conditions?

Methods

Line 151: "read-trimming".

Line 159: For rlog-transformed data did you use DSeq2? If so, this needs to be mentioned. Also, log2. Write "2"in log2 as sub index.

169: mention false-discovery-rate before FDF. And it should be FDR < 0.05.

Line 187: italicize "p" in p-values.

Results

Line 210: "males in the Low Infection group".

Fig. 1: It is important that you also highlight the puncture group, as it is a group that had significant differential expression. You have enough space to spell out the full name of the treatment groups, not just the abbreviation. It would be nice if you added vertical and horizontal axes that intercept 0. It is easier to see the variance in PCs from the center. You could also wrap around elipsoides or hull polygons to the male and female samples. It would be easier to see the sex differences that way.

Line 224: Good. I was just about to ask that.

I wonder how much of the gene expression signal is related to stress due to manipulation.

Line 249: "log2-fold changes".

Fig 3B is not referenced anywhere in the text. If it is not relevant send it to the supplementary data.

Fig. 2, legend: it should be alpha < 0.05, not alpha = 0.05. You will probably need to correct this throughout the manuscript.

Line 274: "log2-fold change".

Line 276: "log2-fold change".

What about genes with a sex-by-immune response interaction? (i.e. sex:treatment)

Discussion

Line 317: Again, I don't think this is too surprising (see my comment on the abstract). However, it is intriguing that you have this level of differentiation between sexes in blood tissue, as this is not a sexually differentiated trait. Perhaps this is something you can discuss in the light of significant evidence and theory explaining sex-biased expression.

Line 235: How does the results presented here directly aid future management and conservation practices? I don't see the connection. How does these kind of studies aid treatment of M. agassizii? I think it is interesting from the scientific point of view, but it is hard to see practical applications.

Lines 329-331: This or a version of this should be in the intro.

Line 338: "studies on turtles"?

Line 341: "In our data,".

Line 353: This should also be in the intro. Expectations about experiments, and why.

Line 361: What do you mean with "previously described"? In another paper or in the results section? Clarify.

Line 263: "log2-fold change".

Line 364: "immunity-related differentially expressed genes".

Lines 379-381: This is ambiguous. Are you referring to your own results or the results of other studies?

Lines 385: Citation?

Line 388: "increasingly starting to be recognized"?

Lines 392-393: This should be in the intro as part of your expectations.

I wonder to what extent hormones play a role in triggering different sex-biased transcriptomic profiles. For example, differences in (mammal) prostaglandin levels can probably trigger sex-biased responses to inflammation and healing.

Line 433: "In this study, ".

Line 434: "GO" instead of "Gene Ontology". Just to be consistent.

Line 435: "tortoises, as expected.".

Line 437: What about sex-specific seasonal variation? You can imagine that as you approach reproductive season hormone levels in males might be geared towards finding mate and perhaps male-male competition, whereas in in females the would be geared towards nesting and resource allocations. Just an idea.

Reviewer #2: This is a novel and interesting use of RNA sequencing to understand differences in diseased and healthy tortoises. Therefore, I commend acceptance, but I suggest some additional references to put this study into a more accurate ecological context, and one important caveat of interpretations in the Discussion.

The main caveat of interpreting this study is that we now know that most tortoises with M. agassizii infections, have very low loads, with no signs of disease, and no adaptive antibody responses. In particular, recent work (Braun et al. 2014, Sandmeier et al. 2017) showed that many animals positive by qPCR for M. agassizii, but do not have a positive antibody response. Similarly, Weitzman et al. (2017) and Sandmeier et al. (2018) showed that many tortoises tested weakly positive for M. agassizii, without showing any signs of URTD. Aiello et al. (2016, 2018) also confirmed that the load of M. agassizii appears to be associated with signs of URTD and adaptive antibody responses, and that adaptive immune responses can take weeks-months to be induced. Therefore, animals with induced antibodies have been experiencing infections over a significant time-frame.

Therefore, I do not think this study was set up to detect differences in innate immunity, which would control such low-load infections. Innate immunity needs to be overcome, in a sense, to stimulate and adaptive antibody response. Therefore, even the “low” infection group has been diseased for a long time and already stimulated the adaptive immune system in the form of induced antibodies. In natural populations, prevalence of induced antibodies in Mojave tortoises is actually relatively low (Sandmeier et al. 2013).

Obviously, it would be hard/impossible to add a whole other group of tortoises to this study, but it should be addressed that even the “low infection” group in fact presents with much higher morbidity than the average naturally-occurring tortoises with low loads of M. agassizii and no signs of morbidity/adaptive immune responses. Calling groups Medium and Severe Infection, and stating in the introduction that no carrier/subclinical/low infection animals were used would be more accurate. Because many of these studies came out after your samples were collected, I see no fault with the conceptualized study design at the time, but only in retrospect!

In particular, I suggest the following changes:

Comments to authors

60: M. agassizii is not known to be intracellular – and the reference (Jacobson et al. 2014) does not claim that this is well understood

66: We know M. agassizii is associated with the mucosal epithelium: we do not know how/if the microbe binds to cells. Many respiratory microbes also are just found within mucosal layer, and not bound to epithelial cells

73: Add “adaptive and” innate immune response (why just the innate response?) Again, I think the study design is not ideal to detecting changes in innate responses during very early/low load infections.

82: Here and thoughout: change “Mycoplasmal” to lower case – “mycoplasma” is a generic term that refers to bacteria within the Mollicutes. It should only be capitalized when it is used as a genus name. Throughout: do not use Mycoplasma to refer to M. agassizii.

334: Reference 1 is listed twice

345-347: This study followed turtles after an experimental infection. Therefore, the investigators likely caught that early, innate response which your study likely missed.

364: Reword to “differentially expressed immune genes”

416-418: I have not seen substantial evidence of sex-biased susceptibility to M. agassizii. In fact, other studies have failed to detect differences (Sandmeier et al. 2013). An alternate interpretation is that sex-biased differences change seasonally – which is also the case for levels of immune cells and functions in tortoises and other reptiles (e.g. Zimmerman et al. 2010, Sandmeier et al. 2016, Goessling et al. 2017).

426-428: These statements seem irrelevant. In fact, the mechanism described cannot occur in animals without sex chromosomes! Make a clearer statement and suggest that sex hormones are likely the driving mechanism of sex differences.

439-440: This statement needs a reference. Much of this behavior starts in April/May in Mojave desert tortoise populations.

455-456: However, females also invest a large amount of energy in producing eggs – female energy expenditure in producing offspring is often downplayed in the literature, but should not be! At this time of year, most females would have already laid eggs. Possibly, they are just depleted of iron?

480: It is incorrect to say that late summer is the “mating season” of desert tortoises.

Additional references (need not all be cited, but provide additional information on the epidemiology of URTD in natural/seminatural populations):

Aiello CM, TC Esque, KE Nussear, PG Emblidge, JP Hudson. 2018. The slow dynamics of mycoplasma infections in a tortoise host reveal heterogeneity pertinent to pathogen transmission and monitoring. Epidemiology and Infection 147:e12.

Braun, J., M. Schrenzel, C. Witte, L. Gokool, J. Burchell, and B. Rideout. 2014. Molecular methods to detect Mycoplasma spp. and testudinid herpesvirus 2 in desert tortoises (Gopherus agassizii) and implication for disease management. Journal of Wildlife Diseases 50:757-766.

Goessling, JM, SA Koler, BD Overman, E Hiltbold, C Guyer, MT Mendoca. 2017. Lag of Immunity across seasonal acclimation states in gopher tortoises (Gopherus polyphemus). J of Exper Zool 327:235-42.

Sandmeier FC, Horn KR, Tracy CR. 2016. Temperature-independent, seasonal fluctuations in immune function of the Mojave desert tortoise (Gopherus agassizii). Canadian Journal of Zoology 94:583-590.

Sandmeier, F.C., C.R. Tracy, B.E. Hagerty, S. DuPré, H. Mohammadpour, and K. Hunter, Jr. 2013. Mycoplasmal upper respiratory tract disease across the range of the threatened Mojave desert tortoise: associations with thermal regime and natural antibodies. Ecohealth 10:63-71.

Sandmeier FC, Weitzman CL, Tracy CR. 2018. An ecoimmunological approach to disease in tortoises reveals the importance of lymphocytes. Ecosphere. 9:e02427.

Sandmeier FC, Weitzman CL, Maloney KN, Tracy CR, Nieto N, Teglas MB, Hunter KW, DuPré S, Gienger CM, Tuma MW. 2017. Comparison of current methods for the detection of chronic mycoplasmal URD in wild populations of the Mojave desert tortoise (Gopherus agassizii). Journal of Wildlife Diseases 53:91-101.

Weitzman, C.L., F.C. Sandmeier, and C.R. Tracy. 2017b. Prevalence of the upper respiratory pathogen Mycoplasma agassizii in Mojave desert tortoises (Gopherus agassizii). Herpetologica 73:113-120.

6. PLOS authors have the option to publish the peer review history of their article (what does this mean?). If published, this will include your full peer review and any attached files.

Reviewer #1: Yes: Santiago Sánchez-Ramírez

Reviewer #2: No

---

## [Author Response · Author response to Decision Letter 0]

22 Jul 2020

Thank you for your time and editorial suggestions to improve the clarity and content of this manuscript. We incorporated your suggestions and reworded sentence structure and supporting information as needed. We have addressed your specific comments and questions in the attached reviewer response document.

Sincerely, 

Cindy Xu, Greer A. Dolby, K. Kristina Drake, Todd C. Esque, and Kenro Kusumi

---

## [Decision Letter · Decision Letter 1]

12 Aug 2020

Immune and sex-biased gene expression in the threatened Mojave desert tortoise, Gopherus agassizii

PONE-D-20-12290R1

Dear Dr. Kusumi,

We’re pleased to inform you that your manuscript has been judged scientifically suitable for publication and will be formally accepted for publication once it meets all outstanding technical requirements.

Kind regards,

Y-h. Taguchi, Dr. Sci.

Academic Editor

PLOS ONE

Additional Editor Comments (optional):

Congratulations! Your manuscript was accepted for the publication in PLoS ONE. Thanks for your submitting your valuable work to our jouranl.

Reviewers' comments:

Reviewer's Responses to Questions

**Comments to the Author**

1. If the authors have adequately addressed your comments raised in a previous round of review and you feel that this manuscript is now acceptable for publication, you may indicate that here to bypass the “Comments to the Author” section, enter your conflict of interest statement in the “Confidential to Editor” section, and submit your "Accept" recommendation.

Reviewer #1: All comments have been addressed

Reviewer #2: All comments have been addressed

2. Is the manuscript technically sound, and do the data support the conclusions?

Reviewer #1: Yes

Reviewer #2: (No Response)

3. Has the statistical analysis been performed appropriately and rigorously? 

Reviewer #1: Yes

Reviewer #2: (No Response)

4. Have the authors made all data underlying the findings in their manuscript fully available?

Reviewer #1: Yes

Reviewer #2: (No Response)

5. Is the manuscript presented in an intelligible fashion and written in standard English?

Reviewer #1: Yes

Reviewer #2: (No Response)

6. Review Comments to the Author

Reviewer #1: (No Response)

Reviewer #2: (No Response)

7. PLOS authors have the option to publish the peer review history of their article (what does this mean?). If published, this will include your full peer review and any attached files.

Reviewer #1: **Yes: **Santiago Sanchez-Ramirez

Reviewer #2: No

---

## [Editor Report · Acceptance letter]

17 Aug 2020

PONE-D-20-12290R1 

Immune and sex-biased gene expression in the threatened Mojave desert tortoise, *Gopherus agassizii*

Dear Dr. Kusumi:

I'm pleased to inform you that your manuscript has been deemed suitable for publication in PLOS ONE. Congratulations! Your manuscript is now with our production department. 

Kind regards, 

on behalf of

Professor Y-h. Taguchi 

Academic Editor

PLOS ONE